# The Shape of the Nasal Cavity and Adaptations to Sniffing in the Dog (*Canis familiaris*) Compared to Other Domesticated Mammals: A Review Article

**DOI:** 10.3390/ani12040517

**Published:** 2022-02-19

**Authors:** Anna Buzek, Katarzyna Serwańska-Leja, Anita Zaworska-Zakrzewska, Małgorzata Kasprowicz-Potocka

**Affiliations:** 1Department of Animal Nutrition, Faculty of Veterinary Medicine and Animal Sciences, Poznan University of Life Sciences, Wołyńska 33, 60-637 Poznań, Poland; anna.buzek@up.poznan.pl (A.B.); anita.zaworska-zakrzewska@up.poznan.pl (A.Z.-Z.); malgorzata.potocka@up.poznan.pl (M.K.-P.); 2Department of Animal Anatomy, Faculty of Veterinary Medicine and Animal Sciences, Poznan University of Life Sciences, Wojska Polskiego 71c, 60-625 Poznań, Poland

**Keywords:** dogs, anatomy, nasal cavity, nasal conchae, brachycephalic syndrome, olfactory system

## Abstract

**Simple Summary:**

Modern dogs are the most morphologically diverse group of mammals. Numerous studies have shown that dogs and their olfactory abilities are of great importance to humans. It was found that the changes within the skull have had a significant impact on the structure of the initial sections of the respiratory system, including the turbinates of the nasal passages. Congenital defects, defined as brachycephalic syndrome, are currently a significant problem in dog breeding as they can affect animal welfare and even lead to death; therefore this manuscript also focuses on this topic. After identifying the subject, we believe that further characteristics of the nasal turbinates and in-depth research related to their hypertrophy are needed. The study also summarized the state of knowledge about the structure of the respiratory system and nasal cavity in dogs compared to other species of domestic animals.

**Abstract:**

Dogs are a good starting point for the description and anatomical analysis of turbinates of the nose. This work aimed at summing up the state of knowledge on the shape of the nasal cavity and airflow in these domestic animals and dealt with the brachycephalic syndrome (BOAS) and anatomical changes in the initial airway area in dogs with a short and widened skull. As a result of artificial selection and breeding concepts, the dog population grew very quickly. Modern dog breeds are characterized by a great variety of their anatomical shape. Craniological changes also had a significant impact on the structure and physiology of the respiratory system in mammals. The shape of the nasal cavity is particularly distinctive in dogs. Numerous studies have established that dogs and their olfactory ability are of great importance in searching for lost people, detecting explosives or drugs as well as signaling disease in the human body. The manuscript describes the structure of the initial part of the respiratory system, including the nasal turbinates, and compares representatives of various animal species. It provides information on the anatomy of brachycephalic dogs and BOAS. The studies suggest that further characterization and studies of nasal turbinates and their hypertrophy are important.

## 1. Introduction

As a result of artificial selection and breeding concepts, the dog population grew very quickly, and these animals began to play a significant role in people’s lives. Modern dog breeds are characterized by a great variety of their anatomical shapes and in effect exhibit various capabilities and perform many specific tasks for humans [1]. This diversity makes the domestic dog the most morphologically diversified mammal species worldwide. This fact is also true with regard to the dog’s skull shape, including the brain [2,3].

This process changed the relationship between humans and animals, leading to a new pressure on selection, absent in the case of the wolf as a protoplast of the domestic dog, which determines the animals’ response to humans and the human environment. In all domesticated animals, including predators, a decreased level of anxiety or fear in contact with humans has become the most important behavioral response change. In effect, animals exhibit a decreased level of responsiveness to external stimuli [4]. The most significant results of these changes in domesticated animals include modifications in the form and size, morphology and functions of the encephalon. Numerous studies have indicated a systematic decrease in the total size of the encephalon in domesticated animals as compared to their wild protoplasts [5,6,7,8]. In contemporary dogs, their encephalon is almost 30% smaller in comparison with that of the wolf. This reduction in size is particularly evident in the limbic system, a part of the brain partly responsible for the fight-or-flight response [9,10]. The skull shape also transformed changed in response to changes in the morphology of the encephalon. Evidence to document this thesis was provided by a well-known experiment conducted in Russia, where the process of silver fox domestication was studied for over 40 years. The study found that changes in the skull shape were correlated with the level of domestication in foxes [11].

Craniological changes also had a significant impact on the structure and physiology of the respiratory system, including turbinates in dogs and other domesticated mammals [12]. The shape of the nasal cavity is particularly significant, as many dog breeds serve humans because of the dog’s fine sense of smell. In recent decades, many scent detection studies have been performed with human, animal and electronic noses. Scent detection by animals has been addressed in studies, which all suggest similar or even superior accuracy compared to standard diagnostic methods [13]. These factors make the dog a highly sensitive real-time olfactive detector.

The highly sensitive sense of smell in dogs is based on several aspects. First, the dog’s ability to detect scents relies on its acute sensitivity when distinguishing individual scents. Secondly, for the dog, even a very low concentration of some substances in the air is sufficient to detect scents. Finally, the dog distinguishes a wider spectrum of scents than is the case with other mammalian species. In other words, dogs are able to analyze more chemical substances present in the air than most other animals [14].

Such an ability is closely linked to anatomical features of the nasal cavity, although, on the other hand, it needs to be remembered that other parts of the skull are also indirectly linked to the sense of smell. This refers in particular to those parts of the neurocranium in which the olfactory pathways of the cerebrum are located. That is why dogs, as the most morphologically diverse mammals, can be a good starting point for the description and anatomical analysis of such complex structures as the nasal turbinates.

This work aimed at summing up the state of knowledge on the shape of the nasal cavity and airflow in the dog and other domestic animal species. In addition, the nomenclature was confronted and descriptions of the nasal cavity organs were compared in view of the occasional inconsistencies in the terminology applied in the literature. Moreover, the work dealt with the brachycephalic syndrome and anatomical changes in the initial airway area in dogs with short mouths.

## 2. Functional Anatomy of Smell

From the very beginning probably, the extraordinary olfactory capacity of dogs has been used by humans to identify and recognize odors [15]. With their approx. 300 million olfactory receptors (humans only have 5–6 million), dogs can detect smells that seem unfathomable to humans. Dogs have olfactory receptors much more sensitive than humans, which is why they are attracted to new and interesting odors (neophilia) [16]. It has been proven that their ability to detect odorants is as much as 10,000–100,000 times that of the average human, with the lower limit of detection in canines being one part per trillion [17].

The important elements of the olfactory system include the nasal cavity, the turbinates covered with the olfactory epithelium with receptors and the vomeronasal organ (VNO) [18], as shown in Figure 1.

The turbinates contribute to an increase in the area lined by the mucosa. However, its total surface may be influenced by the size and shape of the dog’s mouth [19]. The nasal turbinates protrude from the side walls of the nasal cavity and contain the venous network; thus 5–15% of air inhaled by dogs is redirected to them [20].

As air is inhaled, it first enters the nasal turbinates, where a small number of olfactory receptor neurons are located, then flows into the ethmoid turbinates and paranasal sinuses and subsequently toward the pharynx and larynx [21]. The turbinate obstruction alters the way air flows into the olfactory fissure, which in turn affects the sense of smell. The swelling of the turbinate epithelium decreases with exercise, hypercapnia and increased sympathetic tone, while it increases with exposure to cold air and chemical irritants, as well as hypocapnia and increased parasympathetic tone [22].

The VNO (also called Jacobson’s organ) lies along the right-cephalic side of the nasal septum, is symmetrical on both sides and functions as an additional site for odor detection [23,24]. The VNO sensory neurons detect chemical signals that cause behavioral and/or physiological changes [25]. Dennis’s observations in 2003 were expanded and confronted by Salazar et al. in a study of the morphology and physiology of the vomeronasal system in dogs. The VNO of dogs contains both types of epithelia, i.e., receptor and non-receptor epithelium, which are structurally different in the types of nerve fibers and integrated cells [26]. The function of the VNO is to detect a wide range of chemicals including fragrances, in particular pheromones [27,28], and it is believed to play an important role in social behavior and reproduction in dogs. Studies indicate that MRI is the most appropriate imaging technique for VNO visualization, allowing for precise evaluation of soft tissues. This seems important as it has been shown that disturbances in VNO structure may influence behavioral changes in animals. It has been disclosed that VNO sensory epithelial inflammation may show a correlation with the occurrence of intraspecific aggression [29]. In addition, studies on cats indicate a possible association of VNO epithelial inflammation with intraspecific aggressive behavior. Inflammation can weaken the functionality of the epithelium, causing changes in intra-species communication, possibly by reducing the action and perception of chemical communication, which significantly affects behavioral disturbances in cats [30].

An interesting behavior related directly to the VNO is the flehmen reflex. Some species of animals, when taking pheromones from the environment, show a characteristic behavior, the so-called flehmen reflex, involving the curl of the upper lip and the elevation of the nostrils, especially well expressed in horses. Such a reflex improves the movement of inhaled pheromones toward the olfactory bulb [31]. The typical flehmen reflex is not observed in dogs and cats because their upper lips are too rigid and firmly fixed via the frenulum to permit this type of movement. These animal species exhibit different attitudes of behavior, namely they assume a position with an upright head and neck, which they stretch forward for a short time [32]. In cats, an open mouth is observed, then the cat licks the nose area and stares motionlessly into the distance, while the lip remains raised all the time [33]. In dogs, there is a rapid retraction of the tongue toward the incisal papilla which probably also aids the perception of pheromones [32].

Compared to humans, dogs have a much larger olfactory area with an olfactory epithelium that can recognize many more odors. The sensory area occupies more than 30% of the nasal cavity in dogs [34]. Therefore, different breeds of dogs are successfully used for comparative research in olfaction. Not only are they particularly sensitive to fragrances [17], they also have the ability to localize the odor perfectly, even in the presence of a significant background odor, which is possibly thanks to the larger size of their nasal cavity compared to other species [34] and a unique airflow path affected by sniffing. The ability to find the source of the odor, even in the presence of competing odors, makes the tracking dog a key partner in many military operations and police work, e.g., in search and rescue operations [35]. Numerous studies have established that dogs and their olfactory ability are of great importance to medicine. Thanks to their special skills, they successfully detect infectious diseases, neoplasms and metabolic diseases such as diabetes [36]. Recent studies also confirm that dogs can be helpful in detecting COVID-19 [14,37].

## 3. The Nose Structure and Airflow Routes in Animals

The head is the most important and highly specialized part of the body because it contains the brain and vital sense organs related to hearing, sight and smell. In addition to these activities, the nasal cavity also has an extremely important thermoregulatory function. It modifies properties of the inhaled air before it reaches further parts of the respiratory tract. It is heated by passing through the highly vascular surface of the mucosa and moisturized in the process of evaporation of tears and nasal secretions [38].

The cranial cavity is separated from the nasal cavity by a perforated plate called the *lamina cribrosa* ethmoid plate, which is a part of the ethmoid bone (*os ethmoidale*). This bone lies on the border of the facial part of the viscerocranium and the cerebral part, i.e., the neurocranium. The nasal cavity (*cavum nasi*) covers most of the face. In the complex of bones bounding the nasal cavity, the maxilla (*maxilla*) is the largest, which significantly enlarges the initial airways, also forming the maxillary sinus [39]. However, already at this stage, it needs to be stated that in dogs this sinus is minimized only to the maxillary recess (*recessus maxillaris*), limited dorsally by the nasal bone, laterally by the maxillary and incisor bones and ventrally through the palatine process of the maxilla as well as the incisors and the palatine bone. The incisal bone divides the entrance of the nasal cavity and the roof of the palate at the beginning of the skull. In turn, the nasal bone is long and thin, and it is located on the dorsal part of the facial skeleton. Its length may vary depending on the dog breed. The nasal cavity is also divided by the nasal septum (*septum nasi*) into the right and left nasal cavities. This septum is mostly made of cartilage tissue, and it ossifies in the caudal part, forming the vertical plate of the ethmoid bone [40]. The blade, in turn, forms the caudal-ventral part of the nasal septum. The sagittal part is built by two thin lateral layers of bone plates that adhere ventrally and form the cartilaginous part of the nasal septum in the initial part and the vertical plate of the ethmoid bone (the bony part of the nasal septum) [22].

The space within the nasal cavity is significantly limited by the nasal conchae (*conchae nasales*) attached to its walls, which are delicate curved bone structures covered with mucosa that support the sense of smell [40]. The nasal turbinates are characterized by an extremely complicated pattern as indicated in Figure 2. Topographically, we can distinguish the caudal system building the ethmoid labyrinth (*labyrinthus ethmoidalis*), as well as the nasal system, which consists of extensive nasal conchae-dorsal, ventral and middle, being the smallest (Figure 3). The numerous ethmoid turbinates (*ethmoturbinalia*) are separated by ethmoid ducts, which are especially complicated in animals, in which the sense of smell plays an important role [39].

Anatomically we can distinguish the outer ethmoid turbinate (*ectoturbinalia*), the inner ethmoid turbinate, also known as the dorsal turbinate of the nose (*endoturbinale/concha nasalis dorsalis*), and the first, second, third and fourth inner ethmoid turbinates (*endoturbinale I, II, III, IV*) (Figure 4). The first ethmoid turbinate is the most dorsal; it is the longest and farthest in the nasal cavity. It forms the skeleton of the dorsal turbinate. The second ethmoid turbinate is located next to the first and is the skeleton of the middle turbinate. Turbinates III and IV are small in most species. The exception is the dog, in which these turbinates are very well formed. The internal ethmoid turbinates are the medial and dorsal nasal turbinates. The outer turbinates are arranged in one row, with the exception of the horse, where they form two rows [40].

The dorsal nasal turbinate (*concha nasalis dorsalis*) is a curved bone lamina built into the ethmoid crest [41]. It is the longest in comparison to the other turbinates. It extends from the ethmoid plate to the nasal vestibule. Ventral nasal concha (*concha nasalis ventralis*), attached to the jaw through a turbinate crest located in the nasal cavity [42] and the hard palate, models the course of the nasal passages and the distort of the front part of the nasal cavity. The ventral surface of the nasal bone is covered with mucosa, forming the dorsal nasal passage [43]. They are products of delicate tubular bone plates, the shape and structure of which depend on the location or species. In the nasal part, the bone plates do not curl up and are not connected. Caudally, we can see that they curl in such a way that the pinnae touch each other or the side walls of the nasal cavity, thus closing the space belonging to the area of the paranasal sinuses. In the cross-section of the head in the transverse plane, it can be seen that these slits, plates and wires in the cross-section form the letter “E” [39]. The nasal cavities include all the three nasal passages—*meatus nasalis dorsalis*, *meatus nasalis medius* and *meatus nasalis ventralis*—formed by the nasal turbinates [44]. These structures meet near the septum of the nose to form the common nasal duct (*meatus nasalis communis*). The dorsal nasal passages between the vault of the nasal cavity and the dorsal nasal turbinate are observed. They extend to the ethmoid bone and terminate. It is the direct pathway of the inhaled air toward the olfactory region of the nasal cavity. It is also called the olfactory nasal passage, as it terminates in the olfactory region of the nasal cavity. The middle nasal passage is responsible for the connection of the nasal cavity with the paranasal sinuses; thus the term sinus nasal passage is also used. It extends between the dorsal and ventral turbinates and separates in the caudal segment because the middle nasal turbinate is located there. The common (medial) nasal passage lies in the medial plane between the nasal septum and the turbinates, reaching the canopy of the bottom of the nasal cavity. Together with the abdominal (respiratory) duct, they create a path for the air passing toward the nasopharynx [39].

## 4. Comparison of the Nasal Structure in Dogs with Those of Other Domestic Animal Species in Terms of Olfactory Ability

### 4.1. Dogs

There are currently over 368 unique breeds listed by the FCI (Fédération Cynologique Internationale), although several other breeds are also recognized by certain breeders and cynologists. Each of these breeds is a largely genetically isolated population and is described by those organizations that register different dogs as distinct (morphologically and behaviorally) from all other breeds [45]. Pattern variations between breeds are easily recognizable [46]. Many dog breeds are bred specifically to exhibit increased performance in odor detection tasks [47]. As a result, such breeds as German shepherds, terriers and labradors are used as tracking dogs to assist the police or military forces [48]. In contrast, companion dogs, including many brachycephalic breeds, are rarely used for similar tasks [49]. The differences in the olfactory acuity related to brachycephaly are not clearly presented in scientific studies [46]. It has not been unequivocally established whether dogs bred for tracking have a greater olfactory ability than other dogs or wolves [47]. This division, however, raised the question among scientists why some breeds have been originally designed to be utility dogs or tracking dogs, while others have not [49]. One study involving both dogs of different breeds and wolves showed that breeds selected for scent work were superior to brachycephalic breeds and dogs not normally used to detect odors. In the most difficult test, wolves and tracking breeds performed significantly better than the other breeds [14,47]. Morphological differences with regard to smell have also been noted; for example, brachycephalic breeds exhibit olfactory lobe reposition, which may be one of the explanations why they are not popular in detection work [49]. Recent studies by Byrd et al. (2021) suggested that the dog lost olfactory capacity as a result of domestication, and this loss was not recovered in particular breed groupings through directed artificial selection for increased olfactory facility. The dog was shown to have a reduced cribriform plate (CP) area compared to wolves and coyotes and intraspecifically across breeds. Researchers found no significant differences between the CP size of “scent” and “non-scent” breeds [50]. It is commonly believed that the most important feature in detecting dogs is the sharpness of a dog’s sense of smell. However, several authors also point out that dogs used for olfactory or tracking work are now also selected for behavioral reasons. The system of recruiting candidates for police dogs is based mainly on behavioral aspects and the training potential of the dog. Various behavioral, environmental and genetic factors affect the performance of track dogs when used in the police, military or civil service. The use of tracking dogs for operational and tactical purposes is associated with constant cooperation with trainers. Therefore, for the final evaluation of the effectiveness of tracking dogs, not only the sharpness of the dog’s sense of smell is important but also its ability to interact with the trainers. Among the behavioral aspects of a dog’s usefulness as a tracking dog are training, motivation to smell, the ability to focus on searching for and ignoring distracting stimuli, temperament, the willingness to seek without being discouraged by lack of success and the ability to work effectively in stressful situations [51,52,53].

The nasal cavity in dogs is definitely more developed by the ethmoid and nasal turbinates than in other mammals, resulting in the narrowing of the nasal passages in this species [54]. In carnivorous animals, the central part of the turbinate retains the character of the turbinate structure, the nasal part resembles a plate in shape, while the caudal part is slightly thickened. The ventral nasal turbinate has a characteristic shape. It is short and relatively thick, extending from the first premolar (P1) to the third premolar (P3). It begins at the maxillary turbinate crest and then separates into numerous laminae that curl in a spiral. Strongly twisted plaques significantly increase the area covered with the mucosa or the olfactory epithelium. It has numerous recesses. The spiral lamina of the turbinate is covered with a number of secondary laminae, which causes the main recess to have numerous secondary recesses. The middle nasal turbinate is long and equipped with longitudinal folds. Almost the entire caudal part of the nasal cavity is filled with strongly twisted ethmoid turbinates covered with the sensory olfactory epithelium. In the case of dogs, the total area of the ethmoid turbinates by far exceeds the area of the nasal turbinates. The number of ethmoturbinates on each side varies from species to species. The dog has four internal ethmoid and six external ones (Table 1). In some dog breeds, especially hunting or tracking dogs with a strongly developed sense of smell, or generally in macrosomatic animals, the ethmoid turbinates can overgrow the nasal cavity and may also be located in the area of the frontal sinus [39]. However, it is different in brachycephalic dogs. In these animals, morphological changes in the craniofacial area are extensive and lead to frequent anatomical anomalies of the head and neck area [53,55]. In brachycephalic dogs, the base of the skull is shortened, which also includes a reduction in the length of the nasopharynx. This results in disturbances in the structure of the nasal turbinates, which often overgrow [56,57]. All congenital abnormalities, mainly related to the respiratory system, that affect brachycephalic dogs are jointly called the brachycephalic syndrome, which will be discussed later in this publication.

### 4.2. Cats

Dogs often exhibit different anatomical relationships of structures in terms of the wide variation in skull shape. Based on such observations in dogs, it could be anticipated that short-nosed cat breeds also would present some differences in anatomic relationships in the nasal cavity and paranasal sinuses [58]. Brachycephalic cats, similarly to brachycephalic dogs, are very popular nowadays. However, due to their specific, round skull, in which the craniofacial part is shortened, the incidence of comorbidities is high [59]. Cats are carnivorans and as such possess a macrosmatic nasal airway architecture consisting of a dorsal meatus and an olfactory recess. Still, based on their behavior and a comparison of the number of functional olfactory receptor genes in both species, they are thought to have a weaker sense of smell than dogs [60].

This nasal architecture results in a general similarity in the nasal airflow patterns between the bobcat and dog in that the olfactory and respiratory airstreams are separated [35]. The research showed that the structure of the nasal cavity in cats is similar to that in brachycephalic dogs, with the exception that in the cat the ethmoturbinates are larger and the nasal turbinates are correspondingly smaller [58]. The ventral turbinate is small in size, while the middle turbinate is well developed—it reaches the level of the entrance to the maxillary recess [40].

The surface area of the ventral nasal conchae and endo- and ectoturbinals are commonly used as an osteological proxy for the respiratory and the olfactory epithelium, respectively. However, this assumption does not fully account for animals with short snouts, in which these two turbinal structures significantly overlap, potentially placing endo- and ectoturbinals in the path of respiratory airflow. In these species, anterior endo- and ectoturbinals may be covered with the non-sensory (respiratory) epithelium instead of the olfactory epithelium. However, in contrast to those in the dog, anterior extensions of the endo- and ectoturbinals spatially overlap the ventral nasal conchae in both felids (cat and bobcat), and they are covered with respiratory epithelium [61]. In the canine, highly soluble odorants are deposited anteriorly in the sensory region, particularly along the dorsal meatus, while less soluble odorants are deposited more uniformly [62]. The olfactory epithelium in cats is concentrated on the medial aspect of the olfactory recess, with less olfactory epithelium being distributed peripherally; hence, there is less surface area to detect moderately soluble and insoluble odorants. Accordingly, compared with other animals that possess more peripheral olfactory epithelium, cats may have a reduced sense of smell for less soluble odorants [27,63,64,65,66,67].

### 4.3. Horses

The anatomy of the horse’s nose and sinuses is very complex. The nasal cavity is divided into equal halves by the nasal septum and the vomer bone [68]. In horses, the nasal cavity is also filled with nasal turbinates, but they differ in shape from the nasal turbinates of carnivores [54].

Two major nasal conchae in each nasal cavity divide the nasal passage into the dorsal, middle, ventral and common meatus. The nasal passages are paired cavities divided by the nasal septum. These cavities extend for 20–30 cm and are divided into four narrow meatuses by two nasal conchae. The dorsal aspect of the nares leads to the blind-ended nasal diverticulum within the nasoincisive notch [69].

The ventral and dorsal turbinates are built of lamellae twisting in opposite directions—the nave curls toward the middle nasal canal (upward), while the dorsal one curls backward. The caudal part of the nasal turbinate is pneumatized through the nasal maxillary sinus. The caudal part of the dorsal nasal turbinate joins the frontal sinus, forming the conchofrontalis sinus. The nasal parts of the turbinates directly contact the nasal cavity. The medial turbinate is small, but it covers the median sinus [70].

The ethmoid turbinates in horses are small but numerous. As in dogs, they are responsible for increasing the olfactory surface. They are located in the caudal part of the nasal cavity. The inhaled air is transferred to the ethmoid area—to the olfactory epithelium through the dorsal nasal passage. The middle duct enters the paranasal sinuses, while the ventral and common nasal canals are considered to be the main airways. A characteristic feature is the large space, where the two nasal passages connect [39].

The complex paranasal sinus system in horses consists of six pairs of sinuses, with the maxillary and frontal sinuses being of major clinical importance. Sinus compartments communicate with each other, grossly to a large extent creating a rostral and more caudal complex. The rostral complex consists of the ventral conchal sinus, which communicates with the rostral maxillary sinus over the infraorbital canal through the conchomaxillary opening. The caudal complex consists of the caudal maxillary sinus, which broadly communicates with the conchofrontal sinus through the frontomaxillary opening. Over the infraorbital canal, the caudal maxillary sinus also communicates with the more medially located sphenopalatine sinus. The rostral and caudal maxillary sinuses communicate with the nose through separate narrow nasomaxillary openings into the middle meatus [71].

Equines are highly social animals, such as many species of primates, hyenas and dolphins. A few studies have found a well-developed olfactory epithelium in horses, which suggests a large role of the sense of smell [72]. The perception of horses is influenced by factors such as breed, individuality and age.

As in other mammals, the horse’s olfactory organ consists of the olfactory epithelium lining the inside of the upper nasal cavity and connects via olfactory neurons in the turbinates to the olfactory bulb in the horse’s brain. In horses, the tip of the olfactory bulb has been found to coincide with the center of the tuft of hair on the forehead [73]. Horses also have a well-developed VNO which is sensitive to non-volatile and poorly volatile particles often found in body secretions. The rostral end of the incisive duct was blind; thus, unlike most mammals including dogs, there was no communication between the VNO and the oral cavity. Only in the horse were detected nerves in the rostral half of the vomeronasal [74]. When a horse comes into contact with a substance of interest, the molecules related to it activate the VNO which triggers a flehmen reaction, i.e., when the horse arches its upper lip backward and inhales, often with its nostrils closed. The adaptive advantage of the flehmen reaction is that it allows the horse to analyze weakly volatile compounds with much greater accuracy. During the flehmen reaction, the nostrils close, which reduces air leakage and increases the air pressure in the nasal cavity. This allows for the detection of compound molecules by the ploughshare organ [75].

### 4.4. Pigs

Domestic pigs are very intelligent, and they are considered to have a sophisticated olfactory ability [76]. Both wild and domestic pigs use odors for the recognition of ingroup/outgroup differences, status and sexual receptivity [77]. Findings reveal that pig olfactory structures are relatively large, highly organized and follow the general patterns observed in mammals [78]. They have one of the largest olfactory receptor repertoires with 1113 functional olfactory receptor genes and 188 pseudogenes [79,80], which makes pigs legendary for their ability to detect truffles. Trained pigs through their ability to detect and recognize the odorant volatile organic compounds (VOCs) determine the underground locations of truffles [81,82].

Pigs have a very deep but narrow nasal cavity that extends beyond the plane of the eye sockets. Each of the nasal cavities is divided by the pinnacles that make up nasal passages. The post-dorsal nasal passage runs in the area of the nasal cavity, i.e., dorsally in relation to the nasopharynx. This area of the nasal cavity is mainly occupied by the ethmoid turbinate. In the case of pigs, it is fairly well developed, suggesting a good sense of smell, which explains why some pigs are used to look for truffles. From the dorsolateral part of the nasal cavity grows a thick plate, which is the dorsal turbinate of a simple structure, but similarly to ruminants, it is spirally coiled at its end. The superior nasal turbinate is shorter but at the same time more diverse and more complex in structure. It is divided into two secondary laminae, one of which curls upward and the other downward. Thanks to this division, the dorsal part is formed (dorsal recess of the turbinate), while in the lower, frontal part, the abdominal part is formed (the dorsal recess of the turbinate). A sinus is formed in the caudal part of the turbinate [39].

The olfactory areas are quite large and well organized, reflecting the central role that the sense of smell plays in the life of domestic pigs.

### 4.5. Ruminants

The nose in the bovines has an extensive nasolabial plane. The nostrils are comma-shaped and slightly immobile because of the complete cartilaginous skeleton [83].

The size and shape of both endo- and ectoturbinates vary from one individual to another. While there are always four endoturbinates, the number of ectoturbinates is variable. Furthermore, as in humans, bilateral symmetry is often absent [84].

The volume of the nasal cavity in ruminants is smaller than it appears from the outside. The bovine skull is strongly pneumatized by the large frontal sinuses that connect with the nasal cavity [85]. Nasal passage is filled with nasal turbinates. The dorsal turbinate at its end is spiral and covers the sinus of the dorsal turbinate. The nasal turbinate of ruminants is similar to that of a pig. Minor differences appear in small ruminants. In goats, lamellas are thickened, and secondary lamellas grow from the inside of these plaques. In sheep, such a structure occurs only in the lower part of the ventral turbinate. The basal plate of the middle turbinate is divided into two secondary parts—the dorsal plate limits the sinus of the middle turbinate, while the basal plate surrounds the recess of the middle turbinate. In small ruminants, the lamina curls in a spiral, and from the front, it delimits the small ventral sinus of the middle turbinate, and the recess is located more caudally. In ruminants, the nasal septum does not reach the bottom of the nasal cavity and thus forms a single nasopharyngeal passage (meatus nasopharyngeus) [86]. The nasal turbinates divide each nasal cavity into nasal passages—dorsal, median and ventral—which connect to the common nasal passage. The middle turbinate is the largest of all the turbinates. In addition, in ruminants, the ethmoid turbinates lie in the caudal part of the nasal cavity. The ventral nasal passage is the main airway, the dorsal one communicates with the ethmoid ducts, and the middle one connects with the paranasal sinuses [40].

Odors are detected by sensory cells (chemo-receptors) located in the epithelium of the nostrils. However, cattle also possess a second olfactory organ—the Jacobson organ or organum vomeronasale (VNO)—which is located in the mouth in the upper palate and is more sensitive to pheromones than the mucus membrane of the nose.

Research results show that smell is an important sense in ruminants because it complements visual perception, provides information on the social organization in the herd and recognition of individual animals as well as helps to create a bond between the dam and the offspring. The two olfactory systems, i.e., the mucous membrane of the nose and sinus and *organum vomeronasale*, very likely have complementary functions. Olfactory communication between bovines is mainly based on scents or pheromones released but has to be supported by other senses [87].

## 5. Brachycephalia, the Brachycephalic Syndrome and Anatomical Changes in the Nose Balances in Dogs

Considering the large number of dog breeds, we know that morphological differences of these breeds are highly varied and easily discernible. This is one of the reasons why in cynology three types of dog breeds were established: dolichocephalic, mesocephalic and brachycephalic [2]. The term “brachycephalic” means literarily “short, wide-headed” [23]. Brachycephalic breeds are easily recognized by their shortened nose and widely spaced, shallower eye sockets than in the other types of dogs. In dogs, a number of craniofacial anomalies can contribute to brachycephaly, including reduction in bone length, chondrodysplasia of the base of the skull and changes in the position of the palate relative to the base of the skull [88]. According to many studies, these dogs may suffer from the brachycephalic syndrome and respiratory problems, among others.

Brachycephaly is associated with the modification of the skeleton and skull [89]. This results in a distinctive short and very often flattened dorso-ventral snout. This is a result of deliberate efforts by breeders to select dogs for breeding so that they develop local chondrodysplasia and generate individuals with an even more shortened facial skeleton [90,91]. In recent years, there has been a significant increase in the popularity of brachycephalic breeds. According to the British Kennel Club, registrations of French bulldogs increased by 3104%, pugs by 193% and English bulldogs by 96%. Similarly in Australia, according to the ANKC, in the period of 1987–2017, there was an upward trend in the registration of French bulldogs by 11.3% (percentage of the total number of purebred dog registrations), pugs by 320% and English bulldogs by 324% [92]. However, it is impossible not to mention the other side of the coin. In 2014, the Dutch government passed a law prohibiting the breeding of dogs with short mouths with features that may adversely affect the health of dogs and their descendants. Therefore this law applies to all races that have an exaggerated appearance. However, it is not about forbidding the reproduction of selected breeds altogether. Dutch law aims to eliminate from further breeding only those individuals whose physique may cause suffering or serious discomfort to dogs [93].

Despite the growing popularity of these breeds worldwide [55,68,94], it has been shown that the morphological changes are relatively serious and are associated with hereditary head and neck disorders [55,56]. In brachycephalic dogs, the shortened base of the skull also includes a reduction in the length of the nasopharynx [57]. All the congenital anomalies and anatomical abnormalities observed in these breeds, in particular those affecting the respiratory system, but also digestive, eyes, skin and other systems, are described jointly by one term: the brachycephalic syndrome.

The brachycephalic syndrome, also known as the obstructive brachycephalic respiratory syndrome (BAOS or BAS), is a fairly well-described ongoing process of anatomical and functional disorders of the respiratory and digestive systems [94,95]. Dogs diagnosed with BAOS tend to have severe breathing problems as a consequence of the anatomical deformities in the head [96,97]. The basic conditions of the brachycephalic syndrome include congenital anatomical abnormalities such as narrowed nostrils, an elongated soft palate, hypoplastic trachea and nasopharynx and changes in the structure of the nasal turbinates. Figure 5 shows an English bulldog with manifestations of the brachycephalic disorders typical of this type of breed, classified as the brachycephalic syndrome.

The above-mentioned anatomical abnormalities modified the physiology of the respiratory system in brachycephalic breeds [98]. Primary abnormalities such as narrowed nostrils or enlarged turbinates can lead to a significant increase in breathing effort to overcome airway resistance [99]. Chronic airway obstruction in brachycephalic dogs can lead to secondary lesions, including soft palate hypertrophy, laryngeal edema, laryngeal sac degeneration and laryngeal collapse [100].

Clinically, dogs show signs of respiratory distress such as severe dyspnoea, wheezing, coughing, snoring, reluctance to exercise, increased breathing effort, hyperthermia and even collapse [101]. This condition occurs because, despite a marked reduction in the length of the craniofacial skeleton [57], the structures of the soft tissues of the oral cavity and nasal cavity (e.g., soft palate, tongue, tonsils) are not proportionally reduced [102].

In addition to the BAS elements noted above, the authors have observed and documented the presence of the nasal turbinates extending caudally from the nasal cavity to the nasopharynx [44]. The unique anatomy of the skull in the brachycephalic breeds may explain the development of turbinate hypertrophy. The nasal turbinates, along with most of the other bones in the skull, originate from the ectoderm, while the other bones in the body come from the mesoderm. Some bones are formed by ossification on a membranous substrate and tend to form close to the structures they contain [43]. In contrast, cartilage bones continue to grow and ossify only after the end of pregnancy. They are less plastic and tend to grow fully. The nasal turbinates are formed precisely as a result of endochondral ossification, and thus they can grow beyond their immediate vicinity [103,104]. Therefore, in the brachycephalic breeds, the ethmoid turbinates may show a tendency to significantly overgrow into the nasopharynx due to the limited space in the already ossified nasal cavity [44]. The resulting contact between the mucosa-covered turbinates hinders the flow of air through the nose [56]. This is not a normal structure of the initial airway, and studies have shown that turbinate hypertrophy to the pharynx is a common condition in brachycephalic dogs [44]. Clinical cases with hypertrophy and deformation of the nasal turbinates toward the pharynx have been reported many times, while the influence of these anatomical anomalies on the occurrence of BOAS is not fully understood. The most common brachycephalic breeds around the world are pugs, English bulldogs and French bulldogs. Both pugs and French bulldogs are classified as extremely brachycephalic dogs, but there are differences in these breeds concerning clinical signs and skull structure. In pugs, turbinate hypertrophy is a particularly common disease because, despite the shortening of the facial skeleton, it does not shorten the turbinates [105]. It is assumed that turbinate hypertrophy affects up to 21% of these dogs. That is why it is so important to evaluate the nasopharynx for turbinate hypertrophy in all brachycephalic dogs using various methods, including upper respiratory endoscopy [44]. Several other diagnostic methods are also used to assess the anatomical and dynamic changes associated with BOAS, including radiography and computed tomography (CT) [106], while craniometric measurements are also analyzed. In particular, CT allows the obtained data to be reformatted into multiplanar and three-dimensional images. This allows the entire initial airway and throat structures to be visualized. To date, few studies have been published on the CT assessment of the anatomy of the initial airways in brachycephalic dogs, none of which have looked at the measurement of hypertrophied turbinates in specific breeds such as the pug or the bulldog [104,107,108]. These few studies suggest that further characterization and studies of nasal turbinates and their hypertrophy are important. For this purpose, it is worth using computed tomography and three-dimensional imaging tests, which can additionally help characterize the structure of the nasal turbinates and their anatomical anomalies. Research evaluating the prevalence among the breeds as well as the contribution of these structures to the increase in airway resistance and clinical symptoms of the brachycephalic syndrome is warranted. Further anatomical and histological observations in this direction are recommended, which shows that this topic needs to be investigated further, in more detail [44].

## 6. Conclusions

Over the course of thousands of years, dogs exhibited remarkably complex morphological variability. It was influenced by many factors, but one of the most important was the human. Along with the breeding progress, morphological changes also took place. Changes in the skull, in turn, had a significant impact on the structure of the initial sections of the respiratory system, including the turbinates of the nasal passages.

Dogs began to play an important role in the lives of humans, as they were used for hunting, defense, sheep grazing and many other activities. It has been found that some traits are performed better with the help of defense dogs and some with hunting dogs. Numerous studies have shown that nowadays dogs and their olfactory abilities are of great importance in the search for missing people and the detection of explosives or drugs, as well as the detection of diseases in the human body.

Through many years of crossing individuals, new breeds with enhanced specific features were created. Brachycephalic dogs emerged in the same way. Today, these breeds are very popular precisely because of their distinctive appearance. However, attention should be paid to the special nature of the anatomical changes in short-mouthed dogs. Breeding work to shorten the facial skeleton as much as possible in dogs has now reached physiological limits. The breeding of dogs with extreme brachycephaly is currently very controversial, as it is associated with many health problems, including acute and chronic respiratory disorders. The description of the structure and physiology of the nasal turbinates clearly shows how important this structure is for many animal species. At the same time, referring to brachycephaly and the brachycephalic syndrome, we point out how problematic the atypical anatomical changes within them can be.

Extreme brachycephaly is also manifested by conditions unrelated to the respiratory tract, including neurological, dermatological, ophthalmic and orthopedic conditions. As a result, the welfare of the animals is reduced, and their life span is significantly shortened. In summary, dogs with BOAS do not enjoy freedom from discomfort, pain or disease. The growing popularity of these breeds suggests that, unfortunately, health issues are rarely taken into account by prospective owners. Therefore, it is up to veterinarians to ethically advise against breeding dogs with extreme brachycephaly, as these professionals have an ethical obligation to prevent and minimize the negative effects of brachycephaly and hereditary disorders on the health and welfare of dogs.

## Figures and Tables

**Figure 1 animals-12-00517-f001:**
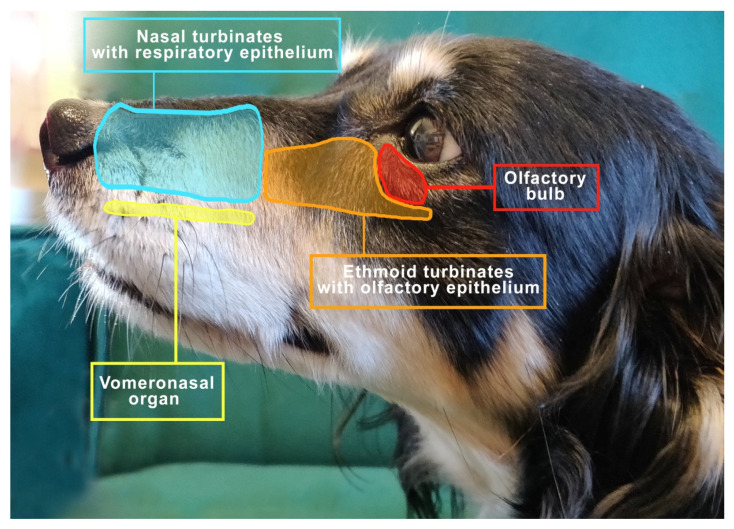
The view to the left of the mesocephalic dog’s head shows the distribution of important structures related to respiration and smell.

**Figure 2 animals-12-00517-f002:**
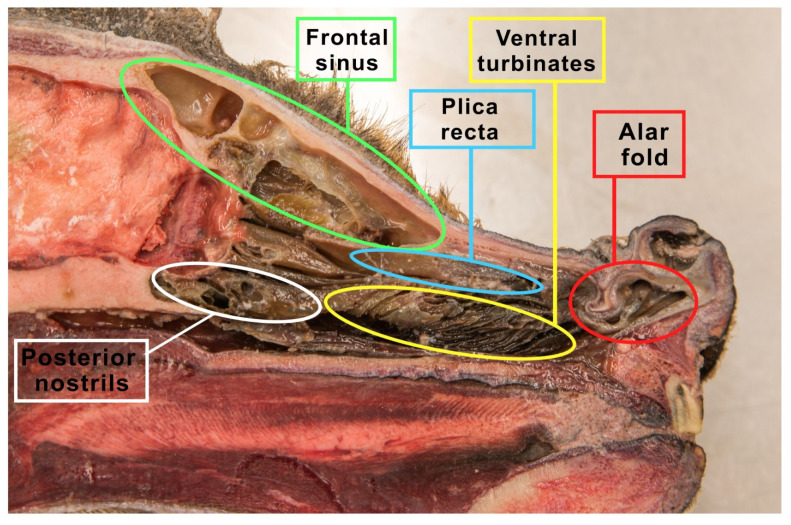
The figure shows a sagittal section of the wolf’s head. The nasal turbinates, the ethmoid labyrinth (made of the inner and outer ethmoid turbinate), nasal passages and the frontal sinus were prepared.

**Figure 3 animals-12-00517-f003:**
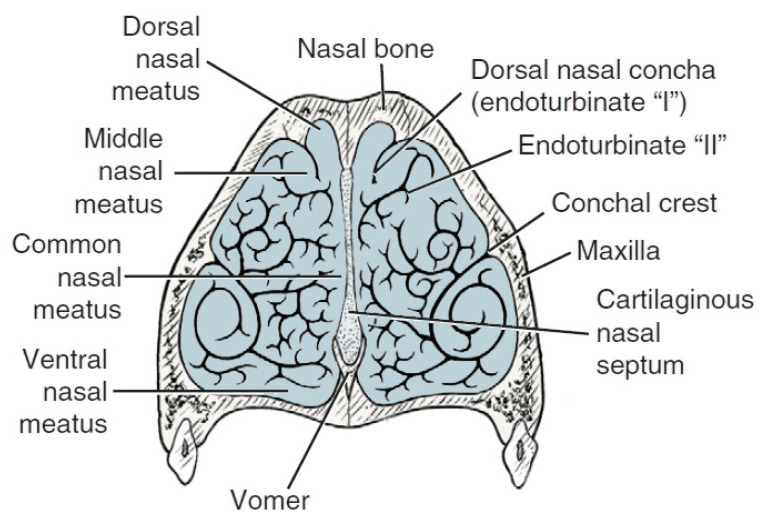
Cross-section of ventral nasal turbinates in dogs. Based on [21].

**Figure 4 animals-12-00517-f004:**
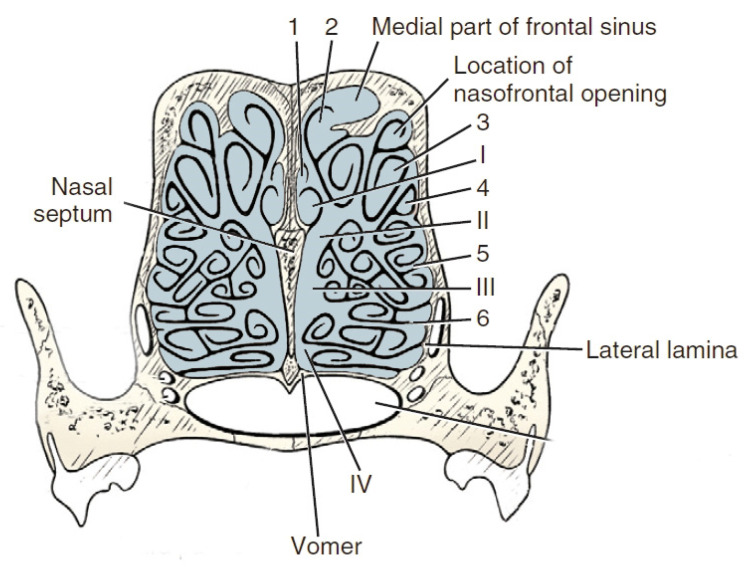
Transverse section of the ethmoturbinates in dogs. I, II, III, IV—endoturbinates; 1, 2, 3, 4—ectoturbinates. Based on [21].

**Figure 5 animals-12-00517-f005:**
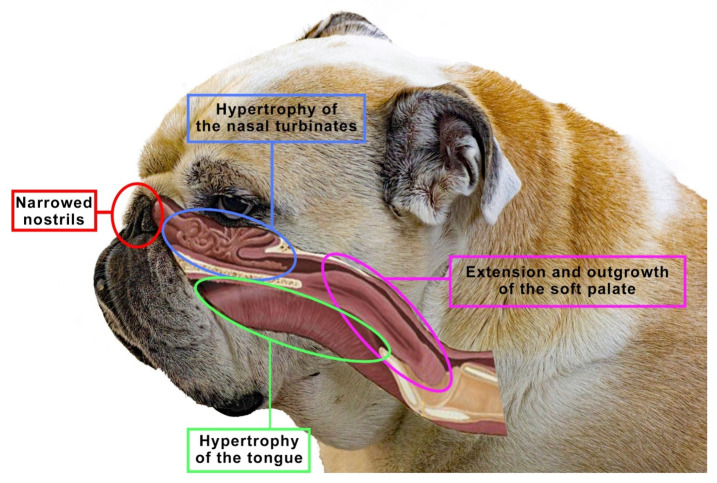
The figure shows an English bulldog. Visible here are signs that are typical for many brachycephalics, classified as the brachycephalic syndrome, such as narrowed nostrils, hypertrophy of the tongue, soft palate or nasal turbinates. Based on [10].

**Table 1 animals-12-00517-t001:** Ethmoid turbinates in domestic animals based on literature [40]).

Item	Ethmoid Turbinates in Domestic Animals				
	Inner Ethmoid Turbinates				Outer Ethmoid Turbinates
	I	II	III	IV	Description
Latin name	*endoturbinale I*	*endoturbinale II*	*endoturbinale III*	*endoturbinale IV*	*ectoturbinale*
Description	The longest; it lies the most dorsally, bone basis for the dorsal turbinate	Bone basis for the middle turbinate	Smaller than turbinates I and II, except for dogs—II, III and IV are particularly well developed in them		short
Dogs	4 inner ethmoid turbinates on one side				6 turbinates on one side
Cats	4 inner ethmoid turbinates on one side				6 turbinates on one side
Horses	6 inner ethmoid turbinates on one side				25 turbinates in 2 rows (one-sided)—external medial and lateral ethmoid turbinates
Pigs	7 inner ethmoid turbinates on one side				20 turbinates on one side
Ruminants	4 inner ethmoid turbinates on one side				18 turbinates on one side

## Data Availability

Not applicable.

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
