# Peer review of "The Shape of the Nasal Cavity and Adaptations to Sniffing in the Dog (Canis familiaris) Compared to Other Domesticated Mammals: A Review Article"

_animals, 2022, doi:10.3390/ani12040517_

Round 1
Reviewer 1 Report
I thank the authors for their tweaks to the paper but still hold my previous view that this review article seems a little confused by what it is trying to achieve. It initially reads as an anatomical description of the dog nose and olfactory function.
In the middle section we have a comparative description of the nose across species and then in conclusion the authors focus on just the brachycephaly which is disjointed. It's long and could be better structured.
I think it would be preferable to have this as a comparative anatomy/ physiology paper in which case much of the text could be replaced by diagrams which would be easier to follow.
Author Response
# Reviewer 1
I thank the authors for their tweaks to the paper but still hold my previous view that this review article seems a little confused by what it is trying to achieve. It initially reads as an anatomical description of the dog nose and olfactory function.
In the middle section we have a comparative description of the nose across species and then in conclusion the authors focus on just the brachycephaly which is disjointed. It's long and could be better structured.
I think it would be preferable to have this as a comparative anatomy/ physiology paper in which case much of the text could be replaced by diagrams which would be easier to follow.
The aim of the publication was to briefly summarize the sense of smell and the anatomical structure of the initial sections of the respiratory system, including the nose. However, taking into account the growing fashion for brachycephalic dogs, it was impossible to avoid the topic of brachycephaly and brachycephalic syndrome, which is a very serious disease entity. By mentioning this subject, we hope that in this way we will reach some breeders and improve the welfare of brachycephalic breeds.
In previous reviews, we tried to correct all errors or inaccuracies indicated by all reviewers, for which we are grateful.
We have added figures showing the cross-section of the nasal cavity in dogs, which will facilitate the reception of the publication.
Reviewer 2 Report
Thank you for offering me to review “The shape of the nasal cavity and adaptations to sniffing in the dog (Canis familiaris) compared to other domesticated mammals: a review article.” The article provides an appropriate overview of the nasal region in the dog and its similarities and differences to other mammalian species.
- - - - Major remarks
The authors refer to the high variability in the terminology of intranasal structures like the turbinal skeleton (lines 86-88), that is mixed throughout the text: for instance, first they talk about the ventral nasal concha (line 216) and the endo- and ectoturbinals (lines 204-213), then they change to the terms maxilloturbinal and frontoethmoturbinals (line 315), respectively, without referring to their homology. This is confusing particularly for readers who are not familiar with the ethmoidal region. The authors should use only one type of terminology. Additionally, a figure for each species illustrating the nasal cavity in cross section and labelling the individual structures will be highly supportive, like for the dog in Evans & de Lahunta (2013, figures 4-27 to 4-29). The figures of the dog's inner nose should be included in the main text, the illustrations of the other species could be considered to be added as supplementary material.
The authors raised the question regarding the choice of specific breeds for odor detection work (line 255f). It has to be mentioned, that breeds and individual dogs, respectively, are also selected according to behavioral reasons. Several authors (see examples below) even emphasize that behavior outweighs smelling ability (“the recruitment system for candidates for police dogs is based mainly on the behavioral aspects and the dog's training potential”, Lesniak et al. 2008:526). This aspect should be referred to in a short section.
- - - Examples for ethological studies referring to working/sniffing dogs:
Adamkiewicz et al. 2013: Traits of drug and explosive detection in dogs of two breeds as evaluated by their handlers and trainers. Anim Sci Rep 31:205-217.
Lesniak et al. 2008: Canine olfactory receptor gene polymorphism and its relation to odor detection performance by sniffer dogs. J Hered 99:518-527.
Svartberg 2002: Shyness-boldness predicts performance in working dogs. Appl Anim Behav Sci 79:157-174.
- - - - Minor remarks:
Line 34: “Moreover, its provide” => it provides
Line 126: “epitheliums” => epithelia
Line 205: “turbinete” => turbinate
- - - - LITERATURE:
Evans HE, de Lahunta A, 2013: Miller's Anatomy of the Dog. 4th ed., Elsevier.
Author Response
# Reviewer 2
Thank you for offering me to review “The shape of the nasal cavity and adaptations to sniffing in the dog (Canis familiaris) compared to other domesticated mammals: a review article.” The article provides an appropriate overview of the nasal region in the dog and its similarities and differences to other mammalian species.
- - - - Major remarks
The authors refer to the high variability in the terminology of intranasal structures like the turbinal skeleton (lines 86-88), that is mixed throughout the text: for instance, first they talk about the ventral nasal concha (line 216) and the endo- and ectoturbinals (lines 204-213), then they change to the terms maxilloturbinal and frontoethmoturbinals (line 315), respectively, without referring to their homology. This is confusing particularly for readers who are not familiar with the ethmoidal region. The authors should use only one type of terminology.
In accordance with the comments of the reviewer, the terminology was standardized
and figures referring to dogs have been added.
The authors raised the question regarding the choice of specific breeds for odor detection work (line 255f). It has to be mentioned, that breeds and individual dogs, respectively, are also selected according to behavioral reasons. Several authors (see examples below) even emphasize that behavior outweighs smelling ability (“the recruitment system for candidates for police dogs is based mainly on the behavioral aspects and the dog's training potential”, Lesniak et al. 2008:526). This aspect should be referred to in a short section.
- - - Examples for ethological studies referring to working/sniffing dogs:
Adamkiewicz et al. 2013: Traits of drug and explosive detection in dogs of two breeds as evaluated by their handlers and trainers. Anim Sci Rep 31:205-217.
Lesniak et al. 2008: Canine olfactory receptor gene polymorphism and its relation to odor detection performance by sniffer dogs. J Hered 99:518-527.
Svartberg 2002: Shyness-boldness predicts performance in working dogs. Appl Anim Behav Sci 79:157-174.
In the added paragraph, the correlation of the severity of smell as well as environmental and behavioral factors in the selection of sniffing dogs was indicated: It is commonly believed that the most important feature in detecting dogs is the sharpness of a dog's sense of smell. However, several authors also point out that dogs used for olfactory or tracking work are now also selected for behavioral reasons. The system of recruiting candidates for police dogs is based mainly on behavioral aspects and the training potential of the dog. Various behavioral, environmental, and genetic factors affect the performance of track dogs when used in the police, military or civil service. The use of tracking dogs for operational and tactical purposes is associated with constant cooperation with trainers. Therefore, for the final evaluation of the effectiveness of tracking dogs, not only the sharpness of the dog's sense of smell is important, but also its ability to interact with the trainers. Among the behavioral aspects of a dog's usefulness as a tracking dog are training, motivation to smell, the ability to focus on searching for and ignoring distracting stimuli, temperament, the willingness to seek without being discouraged by lack of success, and the ability to work effectively in stressful situations. [51-53]
- - - - Minor remarks:
Line 34: “Moreover, its provide” => it provides
Line 126: “epitheliums” => epithelia
Line 205: “turbinete” => turbinate
All mistakes have been corrected.
- - - - LITERATURE:
Evans HE, de Lahunta A, 2013: Miller's Anatomy of the Dog. 4th ed., Elsevier.
Reviewer 3 Report
The review manuscript written by Buzek et al. briefly summarizes the sense of olfaction, nasal morphology, and the associated diseases in domestic dogs. I believe that this review can help non-specialized readers of morphology or chemical sensing to understand the nose of dogs.
I have several very minor comments, so please check the following.
1.
Line 120-126: The sentence “Dennis’s observations in ~ neuroendocrine changes [26]” does not correctly reflect the citation [26]. The reference [26] showed the morphology and histochemical features of the dog VNO, but not the relationship between the VNO and its associated behaviors. I wonder that this sentence was added according to a previous review process. However, it should be deleted and this reference [26] should be cited in the next sentence, that also should be revised as “The VNO of dogs contains both types of epitheliums, i.e.; receptor and non-receptor epithelium, which are structurally different in the types of nerve fibers and integrated cells [26]”.
2.
Line 128, and so on: Throughout the manuscript, “vomeronasal organ” should be “VNO”. I can find some points “vomeronasal organ”. Please check it carefully.
3.
Line 153: What does “around 30% more receptors” mean? According to line 93, dogs have 300 million receptors while humans only have 5 million. If it means the ratio of olfactory area to the nasal cavity, I recommend to describe it as a separated sentence, such as “The sensory area occupies more than 30% of the nasal cavity in dogs”.
4.
Line 374: There is a strong sense of discomfort in the notation “unlike pigs and cows”. This review is mainly about dogs, and the VNO and oral cavity in dogs and cats are also communicated. I recommend to revise this phrase as “unlike most mammals including dogs”.
Author Response
#Reviewer 3
The review manuscript written by Buzek et al. briefly summarizes the sense of olfaction, nasal morphology, and the associated diseases in domestic dogs. I believe that this review can help non-specialized readers of morphology or chemical sensing to understand the nose of dogs.
I have several very minor comments, so please check the following.
1.
Line 120-126: The sentence “Dennis’s observations in ~ neuroendocrine changes [26]” does not correctly reflect the citation [26]. The reference [26] showed the morphology and histochemical features of the dog VNO, but not the relationship between the VNO and its associated behaviors. I wonder that this sentence was added according to a previous review process. However, it should be deleted and this reference [26] should be cited in the next sentence, that also should be revised as “The VNO of dogs contains both types of epitheliums, i.e.; receptor and non-receptor epithelium, which are structurally different in the types of nerve fibers and integrated cells [26]”.
The mistake sentence has been changed by sentence proposed by reviewer.
2.
Line 128, and so on: Throughout the manuscript, “vomeronasal organ” should be “VNO”. I can find some points “vomeronasal organ”. Please check it carefully.
It has been corrected and changed into short form „VNO”.
3.
Line 153: What does “around 30% more receptors” mean? According to line 93, dogs have 300 million receptors while humans only have 5 million. If it means the ratio of olfactory area to the nasal cavity, I recommend to describe it as a separated sentence, such as “The sensory area occupies more than 30% of the nasal cavity in dogs”.
It was indeed about the surface, possibly there was a translation error. I have inserted the proposed sentence.
4.
Line 374: There is a strong sense of discomfort in the notation “unlike pigs and cows”. This review is mainly about dogs, and the VNO and oral cavity in dogs and cats are also communicated. I recommend to revise this phrase as “unlike most mammals including dogs”.
In fact, the tone was weak, so I changed to the suggested sentence.
Round 2
Reviewer 2 Report
General comments:
My suggestions regarding the terminology of anatomical structures and the behavioral aspects for the recruitment of scent dogs have been considered appropriately.
Minor remarks:
Line 125: "epitheliums" => epithelia
Line 125-129: Sentence twice
Line 235: "The dorsal nasal canalies ..." => canal lies?
Author Response
Line 125: "epitheliums" => epithelia – It has been corrected.
Line 125-129: Sentence twice – It has been removed.
Line 235: "The dorsal nasal canalies ..." => canal lies? - It has been changed into passages.
This manuscript is a resubmission of an earlier submission. The following is a list of the peer review reports and author responses from that submission.
Round 1
Reviewer 1 Report
File Attached

Author Response
# Reviewer 1
Comments to the authors:
- You have chosen a very exciting but also incredibly complex topic for your review article. But I think you failed a bit because of the complexity of the topic. When you read the simple summary and the abstract with these many, very diverse set of objectives as well, you do wonder how one could deal with all of this in a readable way in one article. You have put together a lot of information with a lot of diligence, but it would have been better if you had been more limited in your objectives and gone into more detail. There are too many different questions to answer in one article, which is why special features that are very important for the dog have been clearly neglected and only incompletely presented. In the history of the development of dog breeds, you skip 16-20,000 years and start with the search for explosives, drugs and diseases - that is the history of the last decades, before that there are many millennia with completely different breeding goals. They also do not address the shift from breeding dogs for specific abilities to breeding dogs for pure beauty - which then leads to brachycephaly. In describing the development of the skull, much relevant literature on craniometry is missing.
We did not deliberately delve into the history of dog breeds, as we would have to go back several thousand years. This description would dominate the chapter, and that was not our goal.
- In anatomy I unfortunately lost my orientation: you use many terms that do not appear in the nomina anatomica veterinaria, which is very confusing". Some terms I can't place at all, like "auricles" .
We have adapted anatomical terms to the standards in the nomina anatomica veterinaria.
- The illustration contains, in my opinion, very serious anatomical errors
All errors have been corrected. In Figure 1: The toography of nasal turbinates has been correctd; In Figure 2: (a) The "posterior nostrils" has been drawn again, correctly. (b) A "ventral sinus" exactly does not exist in the dog – it has been removed. Also other marks have been taking into account, such as: You have marked the "ventral turbinate" here and also the marking of the "middle turbinates" probably still belongs to the ventral turbinate. c) the dorsal turbinate is the plica recta.
- The thermoregulation of the dog is a very unique mechanism that you are not addressing at all in its specificity. The importance of the lateral nasal gland and its excretory duct, and the peculiarities of thermoregulation during panting - all this can be easily read in scientific papers, but you do not mention it.
The issue of thermoregulation is indeed an interesting topic, but it is impossible to cover all issues related to both the anatomy and physiology of smell. Therefore, the topic of thermoregulation was deliberately omitted by me.
The topic I chose is indeed quite extensive, but I wanted to present, among others, the impact of brachycephaly on anatomical anomalies or changes in the structure of the respiratory system, including turbinates.
Reviewer 2 Report
Generally this review article seems a little confused by what it is trying to achieve. It initially reads as an anatomical description of the dog nose and olfactory function – with the emphasis on the use of the olfactory ability of dogs in human medicine/ work. It then introduces brachycephaly which is a dog welfare issue and due to selective breeding.
In the middle section we have a comparative description of the nose across species and then in conclusion the authors focus on just the brachycephaly.
I think it would be preferable to have this as a comparative anatomy/ physiology paper in which case much of the text could be replaced by diagrams which would be easier to follow. The anatomy descriptions are also rather repetitive and occasionally inprecise which could be avoided by using diagrams.
Specifically
Line 26- short mouth- not precise- use short and widened skull
Lines 147-166 – this whole section could be better described visually using CT or a skull
Figure 2- can the authors please include the auricle which is referred to line 112
Lines 171-195. This section is a detailed anatomical description which again would be better as a diagram.
Lines 193-194. I am not convinced the VNC shapes the nasal cavity. In restricted nasal cavities the VNC distorts in shape……
Line 241-242 – repetitious from above paragraph
Line 262 – throat- can the authors be more precise please- nasopharynx/ pharynx/ neck
Line 400- facial skins?
Line 401- shallower sockets are not proven I think, if so please reference
Line 405-406 – relevance?
Line 409-411- repetition (but better) of 402-403
Line 424- avoid throat
Line 426 – brachycephalic syndrome is not just airway/ digestive but also eye and skin
Fig 3 – the internal anatomy is lifted from the University of Cambridge BOAS website- please acknowledge this. Turbinate anatomy is a French bulldog – not a bulldog
Author Response
# Reviewer 2
- Generally this review article seems a little confused by what it is trying to achieve. It initially reads as an anatomical description of the dog nose and olfactory function – with the emphasis on the use of the olfactory ability of dogs in human medicine/ work. It then introduces brachycephaly which is a dog welfare issue and due to selective breeding. In the middle section we have a comparative description of the nose across species and then in conclusion the authors focus on just the brachycephaly.
I think it would be preferable to have this as a comparative anatomy/ physiology paper in which case much of the text could be replaced by diagrams which would be easier to follow. The anatomy descriptions are also rather repetitive and occasionally inprecise which could be avoided by using diagrams.
While examining brachycephalic dogs, I came to the conclusion that the nasal turbinates would be an interesting topic that could be discussed in the case of brachycephaly. Especially that with brachycephalic syndrome, turbinate hypertrophy and other anatomical anomalies occur. That is why I emphasized this topic quite strongly, and the issue of the impact of brachycephaly on animal welfare, because brachycephaly directly affects, inter alia, on the respiratory system. At the same time, when writing strictly about anatomy, it was impossible not to refer to the air flow path and the physiology of smell.
- Line 26- short mouth- not precise- use short and widened skull –
It has been corrected.
- Figure 2- can the authors please include the auricle which is referred to line 112
Figure 2 has been rebuild.
- Lines 193-194. I am not convinced the VNC shapes the nasal cavity. In restricted nasal cavities the VNC distorts in shape…… -
It has been corrected.
- Line 241-242 – repetitious from above paragraph
It has been corrected.
Line 262 – throat- can the authors be more precise please- nasopharynx/ pharynx/ neck –
It has been corrected.
- Line 400- facial skins?
It was about a shortened nose. It has been corrected.
- Line 401- shallower sockets are not proven I think, if so please reference.
There is a disease syndrome - BOS - Brachycephalic Ocular Syndrome - which describes ophthalmic disorders and complications related to the shortening of the skull. There are also references to the depth of the eye sockets :
https://irishvetjournal.biomedcentral.com/articles/10.1186/s13620-021-00183-5 https://todaysveterinarypractice.com/wpcontent/uploads/sites/4/2016/06/T1503F01.pdf
- Line 409-411- repetition (but better) of 402-403
It has been corrected.
- Line 424- avoid throat
It has been corrected.
- Line 426 – brachycephalic syndrome is not just airway/ digestive but also eye and skin
The lack information has been added.
- Fig 3 – the internal anatomy is lifted from the University of Cambridge BOAS website- please acknowledge this. Turbinate anatomy is a French bulldog – not a bulldog -
It has been replaced by other figure.
Reviewer 3 Report
58- “…limbic system, a nerve structure partly responsible for the fight-or-flight response - is it correct? nerve structure, or part of the brain?
80 In other words, dogs are able to analyze more chemical substances present in the air than other animals [17].
I am not sure if it was proved that dogs are better olfactory detector then wolfs coyotes or for example giant african pouched rats. I haven’t found that statement in cited paper. Dogs probably were trained to detect the biggest number of various substances, but was it compared to other species?
109 The nasal turbinates protrude from the side walls of the nasal cavity and contain the venous net- work, thus 5-15% of air inhaled by dogs is redirected to them.
Since this information is given citation of the mentioned below paper seems to be indicated. Data presented in this paper seems to be worth discussion in manuscript. Publication proposed for citation:
Brent A. Craven, Eric G. Paterson and Gary S. Settles The fluid dynamics of canine olfaction: unique nasal airflow patterns as an explanation of macrosmia Published:09 December 2009https://doi.org/10.1098/rsif.2009.0490
119 The vasoconasal organ (VNO, organ) lies along the right-cephalic side of the nasal septum, is symmetrical on both sides and functions as an additional site for odor detection [23]
The vasoconasal organ- what is this? Shouldn’t it be Vomeronasal ?
In the cited paper [23] authors stated only, that : “Vomeronasal organ (…) is primarily responsible for detecting pheromones.” which don’t have to be odors. However probably the authors statement is correct proper citation should be added.
Regarding to the VNO: It seems to be indicated, during describing the issue of VNO also, to cite at least two other papers of researchers working with this organ: first paper shows ability of evaluation of the morphology of VNO by the fMRI examination in alive dogs and mentioned about possible influence of morphological disorders of this organs on its function. It seems to be indicated, especially taking into account that at the end of the manuscript authors emphasize the importance of research based on the visualization of areas, that are difficult to access during a traditional clinical trial.
Second article that supposed to be worth citation, describes the link between vomeronasalitis in cats and behavioral disorders.
Dzięcioł, M., Podgórski, P., Stańczyk, E., Szumny, A., Woszczyło, M., Pieczewska, B., ... & Wrzosek, M. A. (2020). MRI Features of the Vomeronasal Organ in Dogs (Canis Familiaris). Frontiers in veterinary science, 7, 159.
Asproni, P., Cozzi, A., Verin, R., Lafont-Lecuelle, C., Bienboire-Frosini, C., Poli, A., & Pageat, P. (2016). Pathology and behaviour in feline medicine: investigating the link between vomeronasalitis and aggression. Journal of feline medicine and surgery, 18(12), 997-1002.
Moreover it could be considered by the authors, that due to mentioning the role of VNO it could be indicated to mention specificity of flehmen reflex, and compare it between described in the manuscript species. Morphology of the head, in this case muscle, but also localization of the entrance to this organ, significantly influenced the way, how this reflex is performed in particular species.
Crowell-Davis, S., & Houpt, K. A. (1985). The ontogeny of flehmen in horses. Animal behaviour, 33(3), 739-745.
219-220 There are currently over 368 unique breeds listed by the FCI (Fédération 219 Cynologique Internationale), although several other breeds are also recognized
please format fonts
235 . In the most difficult test, wolves and tracking breeds performed significantly better than the other breeds [39, 42]
[42]- This paper of Hall et al., authors concluded in my opinion, in different way that authors suggested. Hall et al.: ”Our results show that contrary to expectations, Pugs significantly outperformed the German Shepherds in acquiring the odor discrimination and maintaining performance when the odorant concentration was decreased. Nine of 10 Greyhounds did not complete acquisition training because they failed a motivation criterion. These results indicate that Pugs outperformed German Shepherds in the dimensions of olfaction assessed.” Without diminishing the postulated role of the morphology of the broadly understood olfactory organ, the other issues than skull shape seems to be also discussed, since they can play a role in efficiency of olfactory detection.
Suggested position: Kokocińska-Kusiak et al. Canine Olfaction: Physiology, Behavior, and Possibilities for Practical Applications. Anim. 2021, 11 (8), 2463. https://doi.org/10.3390/ani11082463.
The issue of meaning of morphology of the nasal cavity could be discussed in the context of some, maybe existing, other compensatory mechanisms ( e.g. in cats). Evaluation of the efficiency of sense of smell on the basis of only morphology could be misleading. I suppose that could be considered by the authors. We cannot be sure that the dog's model of nasal cave construction is the best and other than morphological features cannot compensate for the olfactory skills in other species.
Horses
Authors describe morphology of the equine nasal cavity, however I am not sure what is the reason – it is not discussed and not compared to the dogs, as it was done regarding the description of the cats nasal cavity. Why authors describe the horse nasal cavity. In the title we can find: adaptations to sniffing in the dog (Canis familiaris) compared to other domesticated mammals: a review article. I see the description but not the comparison. The olfactory detection in horses is not mentioned.
It is the same regarding the pigs. I think that papers of Krzymowski et al. 1999 and other similar seems to be necessary and beneficial to be discussed in this paragraph:
Krzymowski T, Grzegorzewski W, Stefańczyk-Krzymowska S, Skipor J, Wasowska B. Humoral pathway for transfer of the boar pheromone, androstenol, from the nasal mucosa to the brain and hypophysis of gilts. Theriogenology. 1999 Nov;52(7):1225-40. doi: 10.1016/S0093-691X(99)00200-9. PMID: 10735100.
Moreover, description of localization of the entrance to the VNO in all described species seems to be necessary.
Ruminants
In this paragraphs authors decided to mention about olfactory detection, but, first surprise the statement that “smell is an important sense in cattle” . Does it means that it is not important in sheep goats and horses? Authors decided to mention about VNO in cattle – why not in horses, sheep, goats and pigs?
Just an example of the articles describing VNO in some species being animals described in this manuscript
Salazar, I., Quinteiro, P. S., & Cifuentes, J. M. (1995). Comparative anatomy of the vomeronasal cartilage in mammals: mink, cat, dog, pig, cow and horse. Annals of Anatomy-Anatomischer Anzeiger, 177(5), 475-481.
Lee, K. H., Park, C., Kim, J., Moon, C., Ahn, M., & Shin, T. (2016). Histological and lectin histochemical studies of the vomeronasal organ of horses. Tissue and Cell, 48(4), 361-369.
Salazar, I., Quinteiro, P. S., & Cifuentes, J. M. (1997). The soft‐tissue components of the vomeronasal organ in pigs, cows and horses. Anatomia, histologia, embryologia, 26(3), 179-186.
414 In recent years there has been a significant increase in the popularity of brachycephalic breeds
I agree with that constatation, but maybe it would be also worth mentioning, that the Dutch government accepted the law in 2014, which prohibits the breeding of about 20 short-snouted dog breeds, which seems to be important information.
470 e. In pigs ( ?) turbinate hypertrophy is a particularly common disease, because despite the shortening of the facial skeleton it does not 471 shorten the turbinates [88].
I suppose it should be “pugs”
486 Research evaluating the prevalence and prevalence among the breeds. Is that correct?
491 Conclusion
I am not sure if repeating the information about brachycephalic dogs especially without reference to the issue of canine smell mentioned in the title, is necessary in this form. I would suggest modify this section of the manuscript
532- Regarding references, except suggested above positions, I think that the book of T. Jezierski et. al. is worth citing in this paper in a few places.
Jezierski, T.; Ensminger, J.; Papet, L.E. Canine Olfaction Science and Law: Advances in Forensic Science, Medicine, Conservation, and Environmental Remediation; CRC Press/Taylor & Francis Group: Boca Raton, FL, USA, 2016.
Author Response
Reviewer 3
1.58- “…limbic system, a nerve structure partly responsible for the fight-or-flight response - is it correct? nerve structure, or part of the brain?
It has been corrected.
- In other words, dogs are able to analyze more chemical substances present in the air than other animals [17]. I am not sure if it was proved that dogs are better olfactory detector then wolfs coyotes or for example giant african pouched rats. I haven’t found that statement in cited paper. Dogs probably were trained to detect the biggest number of various substances, but was it compared to other species?
Indeed, there are several species that have a better sense of smell than dogs (polar bears or African elephants), and there are also species that have a sense of smell as well as dogs (skunks, cats - generally carnivora). Dogs can be better trained to work with humans, therefore dogs are used to detect various substances and diseases – corrected.
- 109 The nasal turbinates protrude from the side walls of the nasal cavity and contain the venous net- work, thus 5-15% of air inhaled by dogs is redirected to them.
Since this information is given citation of the mentioned below paper seems to be indicated. Data presented in this paper seems to be worth discussion in manuscript. Publication proposed for citation:
Brent A. Craven, Eric G. Paterson and Gary S. Settles The fluid dynamics of canine olfaction: unique nasal airflow patterns as an explanation of macrosmia Published:09 December 2009https://doi.org/10.1098/rsif.2009.0490
- 119 The vasoconasal organ (VNO, organ) lies along the right-cephalic side of the nasal septum, is symmetrical on both sides and functions as an additional site for odor detection [23]
The vasoconasal organ- what is this? Shouldn’t it be Vomeronasal ?
In the cited paper [23] authors stated only, that : “Vomeronasal organ (…) is primarily responsible for detecting pheromones.” which don’t have to be odors. However probably the authors statement is correct proper citation should be added.
The proper citation has been added: Trotier, D. (2011). Vomeronasal organ and human pheromones. European Annals of Otorhinolaryngology, Head and Neck Diseases, 128(4), 184–190. doi:10.1016/j.anorl.2010.11.008
- Regarding to the VNO: It seems to be indicated, during describing the issue of VNO also, to cite at least two other papers of researchers working with this organ: first paper shows ability of evaluation of the morphology of VNO by the fMRI examination in alive dogs and mentioned about possible influence of morphological disorders of this organs on its function. It seems to be indicated, especially taking into account that at the end of the manuscript authors emphasize the importance of research based on the visualization of areas, that are difficult to access during a traditional clinical trial.
Second article that supposed to be worth citation, describes the link between vomeronasalitis in cats and behavioral disorders.
An excerpt has been added about the relationship between feline VNO epitheliitis and behavioral disturbances, and about morphological assessment of VNO by fMRI. The citations has been added: Dzięcioł, M., Podgórski, P., Stańczyk, E., Szumny, A., Woszczyło, M., Pieczewska, B., … & Wrzosek, M. A. (2020). MRI Features of the Vomeronasal Organ in Dogs (Canis Familiaris). Frontiers in veterinary science, 7, 159. -
Asproni, P., Cozzi, A., Verin, R., Lafont-Lecuelle, C., Bienboire-Frosini, C., Poli, A., & Pageat, P. (2016). Pathology and behaviour in feline medicine: investigating the link between vomeronasalitis and aggression. Journal of feline medicine and surgery, 18(12), 997-1002.
- Moreover it could be considered by the authors, that due to mentioning the role of VNO it could be indicated to mention specificity of flehmen reflex, and compare it between described in the manuscript species. Morphology of the head, in this case muscle, but also localization of the entrance to this organ, significantly influenced the way, how this reflex is performed in particular species.
The citation has been added: Crowell-Davis, S., & Houpt, K. A. (1985). The ontogeny of flehmen in horses. Animal behaviour, 33(3), 739-745.
- 219-220 There are currently over 368 unique breeds listed by the FCI (Fédération 219 Cynologique Internationale), although several other breeds are also recognized
please format fonts
It has been corrected.
- 235 . In the most difficult test, wolves and tracking breeds performed significantly better than the other breeds [39, 42]
[42]- This paper of Hall et al., authors concluded in my opinion, in different way that authors suggested. Hall et al.: ”Our results show that contrary to expectations, Pugs significantly outperformed the German Shepherds in acquiring the odor discrimination and maintaining performance when the odorant concentration was decreased. Nine of 10 Greyhounds did not complete acquisition training because they failed a motivation criterion. These results indicate that Pugs outperformed German Shepherds in the dimensions of olfaction assessed.” Without diminishing the postulated role of the morphology of the broadly understood olfactory organ, the other issues than skull shape seems to be also discussed, since they can play a role in efficiency of olfactory detection.
Suggested position: Kokocińska-Kusiak et al. Canine Olfaction: Physiology, Behavior, and Possibilities for Practical Applications. Anim. 2021, 11 (8), 2463. https://doi.org/10.3390/ani11082463 has been cited.
- The issue of meaning of morphology of the nasal cavity could be discussed in the context of some, maybe existing, other compensatory mechanisms ( e.g. in cats). Evaluation of the efficiency of sense of smell on the basis of only morphology could be misleading. I suppose that could be considered by the authors. We cannot be sure that the dog's model of nasal cave construction is the best and other than morphological features cannot compensate for the olfactory skills in other species.
- The question of the importance of nasal cavity morphology could be discussed in the context of some other compensation mechanisms that may exist (e.g. in cats). It can be misleading to assess the olfactory efficiency on the basis of morphology alone. I suppose the authors might consider that. We cannot be sure that the canine model of the nasal cave structure is the best and, apart from morphological features, it cannot compensate for the olfactory abilities of other species.
I do not believe that the dog is the best adapted species in terms of smell or tracking skills, but due to its adaptability and ability to cooperate with humans, I chose the dog as the main species discussed in this publication. Indeed, cats also have a significantly more sensitive and better sense of smell than humans, but in terms of the number of olfactory receptors, they do not match dogs.
- Horses
Authors describe morphology of the equine nasal cavity, however I am not sure what is the reason – it is not discussed and not compared to the dogs, as it was done regarding the description of the cats nasal cavity. Why authors describe the horse nasal cavity. In the title we can find: adaptations to sniffing in the dog (Canis familiaris) compared to other domesticated mammals: a review article. I see the description but not the comparison. The olfactory detection in horses is not mentioned.
Information on horse odor detection and the flehman reflex has been completed.
- Ruminants
In this paragraphs authors decided to mention about olfactory detection, but, first surprise the statement that “smell is an important sense in cattle” . Does it means that it is not important in sheep goats and horses? Authors decided to mention about VNO in cattle – why not in horses, sheep, goats and pigs? Just an example of the articles describing VNO in some species being animals described in this manuscript
Cattle has been changes into ruminants.
New citations: Salazar, I., Quinteiro, P. S., & Cifuentes, J. M. (1995). Comparative anatomy of the vomeronasal cartilage in mammals: mink, cat, dog, pig, cow and horse. Annals of Anatomy-Anatomischer Anzeiger, 177(5), 475-481.
Lee, K. H., Park, C., Kim, J., Moon, C., Ahn, M., & Shin, T. (2016). Histological and lectin histochemical studies of the vomeronasal organ of horses. Tissue and Cell, 48(4), 361-369.
Salazar, I., Quinteiro, P. S., & Cifuentes, J. M. (1997). The soft‐tissue components of the vomeronasal organ in pigs, cows and horses. Anatomia, histologia, embryologia, 26(3), 179-186.
- 414 In recent years there has been a significant increase in the popularity of brachycephalic breeds
I agree with that constatation, but maybe it would be also worth mentioning, that the Dutch government accepted the law in 2014, which prohibits the breeding of about 20 short-snouted dog breeds, which seems to be important information.
The lack information and proper citations has been added.
- 470 In pigs ( ?) turbinate hypertrophy is a particularly common disease, because despite the shortening of the facial skeleton it does not 471 shorten the turbinates [88].
I suppose it should be “pugs”
It is just a mistake. The word “pugs” is correct.
- 91 Conclusion
I am not sure if repeating the information about brachycephalic dogs especially without reference to the issue of canine smell mentioned in the title, is necessary in this form. I would suggest modify this section of the manuscript
I wanted the conclusion to include information about brachycephaly and to present it as unethical. By describing the structure of the nasal turbinates in different species, I wanted to show how important this structure is. And how problematic it can be when there are anatomical anomalies within brachycephalic animals.
- 532- Regarding references, except suggested above positions, I think that the book of T. Jezierski et. al. is worth citing in this paper in a few places.
Jezierski, T.; Ensminger, J.; Papet, L.E. Canine Olfaction Science and Law: Advances in Forensic Science, Medicine, Conservation, and Environmental Remediation; CRC Press/Taylor & Francis Group: Boca Raton, FL, USA, 2016.
The new, above mentioned paper, has been cited.
Round 2
Reviewer 3 Report
The work was corrected according suggestion, what in my opinion makes it better. I recommend to accept this paper in present form.